# Faunistic Analysis of Calliphoridae and Mesembrinellidae (Diptera: Oestroidea) at Different Stages of Bovine Liver Decomposition in the State of Rio de Janeiro

**DOI:** 10.3390/life13091914

**Published:** 2023-09-15

**Authors:** Mariana dos Passos Nunes, Wellington Thadeu de Alcantara Azevedo, Alexandre Sousa da Silva, Cláudia Soares dos Santos Lessa, Jeronimo Alencar, Valéria Magalhães Aguiar

**Affiliations:** 1Departamento de Microbiologia e Parasitologia, Instituto Biomédico, Centro de Ciências Biológicas e da Saúde, Universidade Federal do Estado do Rio de Janeiro, Rua Frei Caneca, 94, Centro, Rio de Janeiro 20211-040, Brazil; mariana.nunes1996@gmail.com (M.d.P.N.); wellingtontaa@yahoo.com.br (W.T.d.A.A.); lessaclss@gmail.com (C.S.d.S.L.); valerialed@yahoo.com.br (V.M.A.); 2Programa de Pós-Graduação em Ciências Biológicas: Biodiversidade Neotropical, Instituto de Biociências, Universidade Federal do Estado do Rio de Janeiro, Avenida Pasteur, 458, Urca, Rio de Janeiro 22290-240, Brazil; 3Programa de Pós-Graduação em Biologia Animal, Instituto de Biologia e Ciências da Saúde, Universidade Federal Rural do Rio de Janeiro, Estrada Rio-São Paulo, Seropédica 23890-000, Brazil; 4Departamento de Matemática e Estatística, Instituto de Biociências, Centro de Ciências Biológicas e da Saúde, Universidade Federal do Estado do Rio de Janeiro, Avenida Pasteur, 458, Urca, Rio de Janeiro 22290-240, Brazil; alexandre.silva@uniriotec.br; 5Laboratório de Diptera, Fundação Oswaldo Cruz, Avenida Brasil, 4365, Rio de Janeiro 21040-360, Brazil

**Keywords:** diversity, necrophagous insects, blowfly, bioindicator

## Abstract

Performing quantitative sampling and determining faunistic analyses of dipterans is of fundamental importance in the analysis of ecological behavior, such as population dynamics and diversity, among other factors, for exotic and native species of necrophagous dipterans, so it is important to observe the type of bait used in traps to capture these dipteran species. This work aims to study structural parameters and faunistic indices of the diversity of Calliphoridae and Mesembrinellidae species as well as the abundance and diversity of species attracted to liver in two stages of decomposition: fresh liver and liver at 48 h of putrefaction. A total of 2826 dipterans were collected during the period from May 2021 to February 2022. We observed that liver in decomposition for 48 h was more attractive in the forest and rural environments, while fresh liver showed greater attractiveness in the urban environment; however, no statistical difference was evidenced between the attractiveness in the different environments. The Mesembrinellidae family and the species *Lucilia eximia* (Wiedemann, 1819) were collected mostly from deteriorated liver, while *Cochliomyia macellaria* (Fabricius, 1775) showed no preference for any liver decomposition stage. The Wilcoxon test indicated that there is a significant difference between the preferences for putrefied bait in Mesembrinellidae, while in Calliphoridae, there was no preference for type of bait. The faunistic analysis showed that richness in the forest area was always higher when compared to the urban and rural areas. *Laneela nigripes* (Guimarães, 1977) and *Mesembrinella bellardiana* (Aldrich, 1922) were abundant and exclusive in the preserved environment, showing themselves to be good environmental bioindicators.

## 1. Introduction

Among the most abundant dipterans in the state of Rio de Janeiro are those of the Calliphoridae family [1], which have an average size ranging from 4 to 16 mm; popularly known as blowflies, they have a metallic coloration and can be greenish, bluish, violet, and cuprean. They are characterized by being natural colonizers, with a high rate of dispersion and the ability to locate distant resources [2,3]. They are distributed worldwide, except in Antarctica, with approximately 100 genera and over 1500 species [4]; 130 are present in the Neotropical region, of which 10 genera and 29 species are already registered in Brazil [5]. 

Meanwhile, Mesembrinellidae, another relevant family in Rio de Janeiro, has more robust dipterans that are medium to large in size, between 7 and 16 mm, are brown in color, and may present metallic coloration in the abdomen in some species [6]. To date, there are 15 living species distributed in five genera in Brazil [5] and 53 species throughout the Neotropical region [7,8], being abundant in forest environments [1,3]. A peculiarity of this family among Oestroidea is the larval development, which occurs inside the maternal oviduct. Larvae are only released to the external environment when they are about to pupate, when they reach the last instar. This characteristic, where the female does not ovipose, is called adenotrophic viviparity [3].

Dipterans characterize the most varied and abundant organisms present in decomposing organic matter such as fruits, manure, carrion, carcasses, and rotten fungi, among others, playing a fundamental ecological role in the cycling of nutrients [9,10]. The carcass and its associated fluids, though they are temporary resources that deplete themselves at a rate directly proportional to the population they support, can be exploited in multiple ways by several groups of insects simultaneously [11]. For calliphorids, for example, this resource is valuable since the contact of adult females with protein substrate stimulates oogenesis in their ovarian follicles and serves as a source of sustenance and development of immature members of the species [12], and by degrading different types of carcasses, they are indispensable in all terrestrial ecosystems [13].

Dipteran sampling studies are of fundamental importance in analyzing the ecological behavior, such as population dynamics, diversity, distribution, dispersal areas, and seasonality, of exotic and native species of necrophagous dipterans. Thus, it is important to observe the type of bait used in the traps to capture these dipterans species [14]. Several research studies around the world have already been carried out to evaluate the ecological behavior of dipterans of the Calliphoridae family using different types of bait traps; for example, putrefied sardine bait has been shown to be attractive to calliphorids [15], being used in several studies [6,16]. Other bait types used were fish meat and pork liver [17]; chicken viscera and dog feces [18]; and rotten banana and squid [19]. Bovine lung was also used to capture sarcophagids [20], and bovine liver proved proficient in capturing calliphorids [21].

Assessing the diversity, abundance, and other faunal patterns of the species of these families at different types and stages of bait decomposition is important in order to gain broader knowledge of this group, which is recognized for its medical, forensic, and ecological importance. This study aimed to analyze the abundance and richness of Calliphoridae and Mesembrinellidae species attracted by liver in two stages of decomposition, fresh liver and liver at 48 h of putrefaction, evaluating structural parameters and faunal indices and identifying species that can act as environmental bioindicators.

## 2. Materials and Methods

Collections were carried out in four georeferenced sites of three ecological areas evaluated in the state of Rio de Janeiro, as shown in Table 1 and Figure 1, where an attractive bait trap was installed at each point to capture Diptera. These three areas correspond to the Parque Estadual dos Três Picos (PETP), a Conservation Unit located in the Serra do Mar, the mountainous region of the state of Rio de Janeiro, with Scientific Research Authorization in the Conservation Unit of the INEA N° 019/2020; another area was located at Universidade Federal Rural do Rio de Janeiro (UFRRJ), in the municipality of Seropédica; and the third area chosen for this study was the Universidade Federal do Estado do Rio de Janeiro (UNIRIO) at the campus located in the neighborhood of Urca in the city of Rio de Janeiro.

The experiment was conducted between June 2021 and February 2022. In each area, four traps produced through PVC pipes were installed, manufactured according to [22]. The attractive bait was placed at the base of each trap. An amount of 300 g of fresh liver was used as bait for two traps, while liver at 48 h of putrefaction was used in the other two, and each trap was exposed for 48 h.

The collected dipterans were sacrificed in a solution based on ethyl alcohol and ethyl acetate. They were then referred to the Laboratório de Estudos de Diptera (LED-UNIRIO) where they were transferred with their respective identification to petri dishes lined with absorbent paper, sealed with plastic film, and stored in a freezer at 4 °C until sorting. The insects were then identified under incident light using stereoscopic microscopes (Olympus SZX7) following taxonomic keys [23,24]. Insects were affixed and stored in the entomological collection of the National Museum and in the LED-UNIRIO collection.

The following indices were applied as faunal analyses: the richness estimator jackknife 1st order (Sjack1) was used to evaluate the species richness of a given community. The Shannon–Wiener–Wiener diversity index was used to measure biodiversity, taking into account the number of species and the dominant species. Simpson dominance measures the probability that two individuals, randomly selected from the sample, belong to the same species. The Pielou evenness index, derived from the Shannon–Wiener diversity index, was used to represent the uniformity of the distribution of individuals among the existing species.

In order to establish the species as constant, accessory, or accidental at each collection point, the formula for constancy of occurrence (C) C = n × 100/N was applied, where n = number of collections containing the species under study, and N = total number of collections performed. This resulted in specimens being classified as constant (C > 50%), accessory (25% < C < 50%), or accidental (C < 25%) [25]. According to the frequency, classified as rare, intermediate, or common [26], rare species were those that had 1 or 2 individuals collected per locality; intermediate when species were collected with 3 to 51 individuals; and common when 52 or more individuals were collected. All statistics were calculated using R version 4.2.1 and DivEs—Diversity of Species version 4.0.

The Shapiro–Wilk normality test was used to evaluate the abundance variable, and since there was no normality in the data, the use of nonparametric tests was recommended. The Kruskal–Wallis and Wilcoxon tests, which compare independent samples, were used to evaluate the influence of the type of bait used, evaluating the degree of association between the variables, considering a significance level of 5% for the tests [27,28].

## 3. Results

A total of 2826 dipterans were collected during the period from May 2021 to February 2022, and Figure 2 represents the box plot of the abundance of individuals collected from the families Calliphoridae and Mesembrinellidae for each type of preferred bait (fresh bait or putrefying bait); in the three collection areas, forest area (Parque Estadual dos Três Picos—Cachoeiras de Macacu), rural area (UFRRJ—Seropédica), and urban area (UNIRIO, campus Urca—RJ); and in the four seasons (autumn 2021, winter 2021, spring 2021, and summer 2022). The Wilcoxon test showed that there is a statistical difference between the preference for putrefying bait in the Mesembrinellidae family (w = 155.5, *p* = 0.03), while the Calliphoridae family had no significant preference (w = 1134.5, *p* = 0.16).

As for the degree of decomposition of the bait (Table 2), we observed that the liver in decomposition for 48 h was more attractive in the forest and rural environments, while fresh liver presented greater attractiveness in the urban environment. However, there was no statistical difference between the attractiveness in the environments. The Mesembrinellidae family and the species *Lucilia eximia* were collected mostly from deteriorated liver, and *Cochliomyia macellaria* showed no preference for the stage of liver decomposition. Meanwhile, for species that occurred in urban and rural areas, their collection was mostly in fresh liver when comparing these two areas only with each other.

Table 3 and Table 4 show that for the most part, species are constant and intermediate, regarding their constancy and frequency, respectively, in the forest, rural, and urban environments. Calliphoridae presented three species for each degree of constancy, while the Mesembrinellidae species were mostly accidental; furthermore, in relation to frequency, most of the species of both families were intermediate. *H. segmentaria, H. semidiaphana,* and *L. eximia* were constant and common Calliphoridae species in general, while *La. nigripes* and *M. bellardiana* were constant and common Mesembrinellidae species in forest areas.

Table 5 shows that the richness in the forest area is higher during the four seasons when compared with the urban and rural areas, while in these latter areas, the richness assumes similar values. The Shannon–Wiener diversity was higher in the forest environment during the autumn; in winter and spring, the forest and urban areas had similar values; in the summer, the urban area showed greater diversity, followed by the rural and finally the forest area. Simpson dominance was higher in the urban area during the autumn; in the rural area during winter and spring; and in the forest area during the summer. Pielou evenness had approximate values in the forest and rural areas during the autumn and in rural and urban areas during the winter; it was greater in the forest area during the spring and had approximate values in rural and urban areas during the summer.

## 4. Discussion

Because these organisms can be found in natural and/or urban environments [29], it is important to study the faunal aspects within a biome, space, and time. In work carried out at the Botanical Garden of Rio de Janeiro, [30] found greater diversity of Calliphoridae species in less anthropic areas. In this study, the Shannon–Wiener–Wiener diversity index, which states that the higher the value of this index, the greater the uncertainty that a species belongs to the individuals selected in the sample, also varied in the environments throughout the seasons, ranging from 1.49 to 2.22 in the forest environment; 0.66 to 1.83 in the rural environment; and 0.69 to 1.92 in the urban environment. In the Simpson dominance index, the higher the value, the greater the probability of the individuals being of the same species, and in this study, the index was higher in rural and urban environments than in the forest in all seasons of the year. The Pielou evenness index, which represents the distribution of families in an environment, showed higher values in the summer, indicating that the distribution was more abundant in this season, while the lowest values were in the spring.

Through the knowledge obtained of the abundance, diversity, and other faunal patterns of Calliphoridae and Mesembrinellidae, it is possible to point out measures of preservation of environments, mainly using bioindicator species in this context. Calliphoridae is constantly abundantly present in the first minutes of the first phase of active decomposition of carcasses, containing species of high forensic relevance [31]; in the Neotropical region, the following genera are primarily cited: *Chrysomya*, *Lucilia*, *Cochliomyia*, and *Hemilucilia* [32]. Within the medical–forensic field of entomology, each group is adapted to colonize a certain stage of decomposition [33]. Therefore, it is important to know the fauna of insects and arthropods present in different organs, like the liver and other body structures, as well as in different levels of decomposition.

Despite the relatively low knowledge in the literature about the reproductive biology of Mesembrinellidae in relation to Calliphoridae, females of the first family are usually attracted by the exhaled odor of substrates of plant or animal origin in decomposition or fermentation processes; furthermore, the substrate used by the larva to end its development after partially maturing within the female′s abdomen is unknown [8]. However, in the data presented here, although Calliphoridae species in general do not have a preference for the stage of liver decomposition, Mesembrinellidae preferred putrefied liver; this included some species of this family that, even with the low number of samples, were not collected in fresh liver, such as *E. quadrilineata*, *E. cyaneicyncta*, *E. benoisti*, and *H. aneiventris*.

Recent studies attest to the efficacy in the use of bovine liver to capture dipterans within the Calliphoridae family [34], especially in the genus *Cochliomyia* [35]. In this work, the bait in question was attractive in the capture not only of *Cochliomyia* but also of Calliphoridae and Mesembrinellidae in general. However, the low capture of certain species, typical of anthropic areas such as, for example, those belonging to the genus *Chrysomya*, may reveal its lack of attraction to this type of bait or its low temporal abundance in the urban environment. As previously demonstrated, there was a higher abundance of species in putrefied bait; however, this was due to the fact that a numerically high number of specimens were collected in the forest environment when compared to other environments. If we take into account only the urban and rural areas, the number of specimens collected in fresh bait was mostly higher than in putrefied bait, and this is an interesting fact, since in most studies, in most different environments, putrefied bait, such as sardine fish bait, is usually used in the capture of Diptera, especially of the Calliphoridae family [16,36].

Species of the Calliphoridae family that occurred in the most preserved environment appeared in greater abundance in putrefied liver, while these same species in rural and urban environments appeared in greater abundance in fresh liver. *L. eximia* was the predominant species in different collection environments in [37]; similarly, this study presented it as constant and common in the three environments. However, for *L. cuprina*, several studies in the Atlantic Forest, such as [36,38,39], show this species presenting low frequency or being absent; in this work, the species was constant and intermediate in the rural environment and accessory and constant in all three locations.

The genus *Chrysomya* was collected in urban and rural environments, with *C. megacephala* as constant and intermediate in both environments and accessory and intermediate in general. *C. albiceps* was considered accessory and rare in the urban environment, accessory and intermediate in the rural environment, and accidental and intermediate in general, which corroborates the findings of [39], who showed this species preferring anthropic areas. Other studies have shown the presence of this species included in environments close to forest areas in the state of Rio de Janeiro, such as [40,41]. However, in this study, the species only showed distribution in relatively urban areas.

*C. macellaria* was constant and intermediate in the rural environment and accessory and constant in general. Unlike [30], who considered it as intermediate and accidental, however, the frequency of this species only in rural areas is due to the ecological displacement caused by competition with the exotic species of the genus *Chrysomya* that have become dominant in urban areas [42]. Representatives of the genera *Hemilucilia* and *Paralucilia* are, respectively, partially and substantially predominant in preserved areas [40,43], whereas in the present study, *H. segmentaria* and *H. semidiaphana* were common in urban and rural environments; in addition, in this study, it was demonstrated that Mesembrinellidae prefers putrefied bait, and *H. benoisti* and *P. nigrofacialis* also appeared in greater abundance in this type of bait.

*L. nigripes* and *M. bellardiana* were the only constant and common Mesembrinellidae species in the forest environment, and they were accidental and common considering the three collection environments. These data corroborate [41,44,45], where they were considered the most abundant in less anthropic areas. Considering only the forest environment, *M. peregrine* and *Hu. Aneiventris* were accessory and intermediate; *M. semihyalina, M. currani*, and *E. quadrilineata* were constant and intermediate; *E. cyaneicyncta* and *E. benoisti* were accessory and rare; and *E. randa* was constant and rare.

The present study corroborates the data in the literature, since all species of this family were presented in preserved environments; thus, they can be used as good bioindicators of preserved environments, especially the two Mesembrinellidae species that were constant and common in the forest environment, namely, *La. nigripes* and *M. bellardiana*.

We concluded that the richness is higher in the forest environment thanks to the presence of species of Mesembrinellidae; the diversity, in a major way, presents higher indices in the forest environment; finally, for dominance and equity, the indices are close between the environments.

Regarding the abundance and diversity of species attracted by fresh liver and liver at 48 h of putrefaction, it is concluded that the Mesembrinellidae family preferred putrefied bait, while Calliphoridae did not present a significant preference. *L. eximia* occurred most abundantly in the putrefying liver, whereas for most Calliphoridae species, total abundance was slightly higher in fresh bait. This study reinforces the idea that the family Mesembrinellidae can be used as an environmental bioindicator, especially the species *La. Nigripes* and *M. bellardiana,* which occurred in a more abundant, common, constant, and exclusive way in a preserved environment, being totally asynanthropes.

## Figures and Tables

**Figure 1 life-13-01914-f001:**
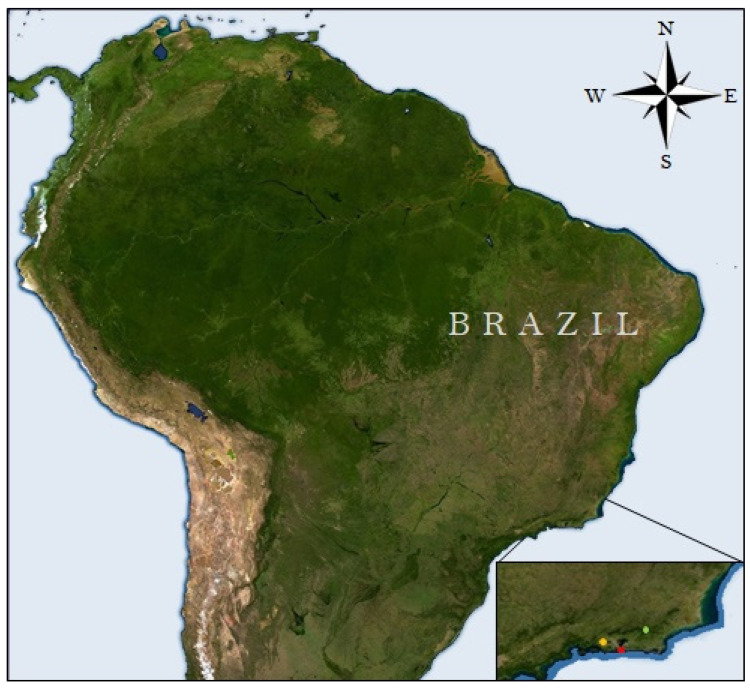
Image of the three ecological areas in the state of Rio de Janeiro where the research was conducted: forest area (demarcated in green color) in the Parque Estadual dos Três Picos; rural area (demarcated in yellow color) including cattle farming on the campus of the Universidade Federal Rural do Rio de Janeiro; and urban area (demarcated in red color) at the campus of the Universidade Federal do Estado do Rio de Janeiro, in the neighborhood of Urca. Source: earth.google.com/web/ (accessed on 7 August 2023).

**Figure 2 life-13-01914-f002:**
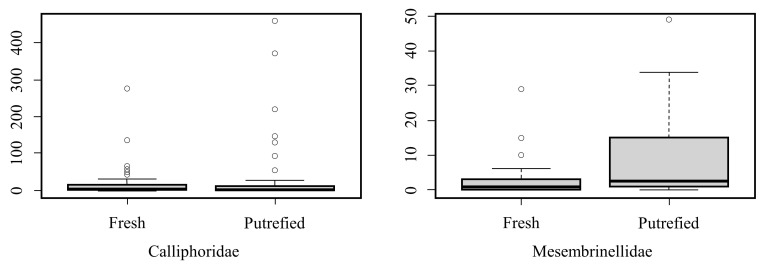
Box plot of the abundance of Calliphoridae and Mesembrinellidae in different degrees of decomposition of bovine liver bait (fresh and at 48 h of putrefaction) in a forest area (Parque Estadual dos Três Picos), rural area (UFRRJ), and urban area (UNIRIO) in the four seasons of the year between 2021 and 2022.

**Table 1 life-13-01914-t001:** Georeferencing of the ecological areas studied in the state of Rio de Janeiro. Forest area—Parque Estadual dos Três Picos; rural area—cattle farming on the campus of the Universidade Federal Rural do Rio de Janeiro; and urban area—campus of the Universidade Federal do Estado do Rio de Janeiro, in the neighborhood of Urca.

Locations	Forest	Rural	Urban
Site 1	22°25′07.13″ S	22°45′40.42″ S 43°42′09.23″ O	22°57′19.42″ S
	42°36′11.97″ O		43°10′08.56″ O
Site 2	22°25′07.75″ S 42°36′11.09″ O	22°45′41.01″ S 43°42′09.84″ O	22°57′18.93″ S
			43°10′10.17″ O
Site 3	22°25′07.06″ S 42°36′09.92″ O	22°45′40.57″ S 43°42′09.67″ O	22°57′16.86″ S
			43°10′08.99″ O
Site 4	22°25′06.17″ S	22°45′39.83″ S 43°42′09.57″ O	22°57′17.30″ S
	42°36′09.80″ O		43°10′10.99″ O

**Table 2 life-13-01914-t002:** Absolute and relative abundance of Calliphoridae and Mesembrinellidae in different degrees of decomposition of beef liver bait (fresh and at 48 h of putrefaction) in a forest area (Parque Estadual dos Três Picos—Cachoeiras de Macacu), rural area (UFRRJ—Seropédica), and urban area (UNIRIO, campus Urca—RJ) in the four seasons of the year between 2021 and 2022.

Species	Forest				Urban				Rural			
	Fresh		Putrefied		Fresh		Putrefied		Fresh		Putrefied	
Calliphoridae	n	%	n	%	N	%	n	%	n	%	n	%
*Chrysomya albiceps* Wiedemann, 1819	-	-	-	-	1	0.29	-	-	-	-	3	7.89
*Chrysomya megacephala* Fabricius, 1794	-	-	-	-	16	4.76	4	8.33	5	4.90	2	5.26
*Cochliomyia macellaria* Fabricius, 1775	-	-	-	-	-	-	-	-	14	13.70	9	23.60
*Hemilucilia benoisti* Séguy, 1925	1	0.18	3	0.16	-	-	-	-	-	-	-	-
*Hemilucilia segmentaria* Fabricius, 1805	51	9.60	290	16.37	210	62.50	4	8.33	5	4.90	2	5.26
*Hemilucilia semidiaphana* Rondani, 1850	47	8.85	160	9.03	47	13.99	-	-	-	-	-	-
*Lucilia cuprina* Wiedemann, 1830	-	-	-	-	-	-	-	-	7	6.86	5	13.16
*Lucilia eximia* Wiedemann, 1819	348	65.54	1078	60.87	62	18.45	40	83.33	71	69.61	17	44.74
*Paralucilia nigrofacialis* Mello, 1969	6	1.13	14	0.79	-	-	-	-	-	-	-	-
Mesembrinellidae												
*Laneela nigripes* Guimarães, 1977	45	8.47	130	7.34	-	-	-	-	-	-	-	-
*Mesembrinella bellardiana* Aldrich, 1922	27	5.08	74	4.18	-	-	-	-	-	-	-	-
*Mesembrinella peregrina* Aldrich, 1922	1	0.18	4	0.22	-	-	-	-	-	-	-	-
*Mesembrinella semihyalina* Mello, 1967	2	0.37	1	0.05	-	-	-	-	-	-	-	-
*Mesembrinella currani* Guimarães, 1977	2	0.37	1	0.05	-	-	-	-	-	-	-	-
*Eumesembrinella quadrilineata* Fabricius, 1805	-	-	8	0.45	-	-	-	-	-	-	-	-
*Eumesembrinella cyaneicyncta* Surcouf, 1919	-	-	2	0.11	-	-	-	-	-	-	-	-
*Eumesembrinella randa* Walker, 1849	1	0.18	2	0.11	-	-	-	-	-	-	-	-
*Eumesembrinella benoisti* Séguy, 1925	-	-	1	0.05	-	-	-	-	-	-	-	-
*Huascaromusca aeneiventris* Wiedemann, 1830	-	-	3	0.16	-	-	-	-	-	-	-	-
Total	531	100	1771	100	336	100	48	100	102	100	38	100

**Table 3 life-13-01914-t003:** Constancy and frequency of Calliphoridae and Mesembrinellidae species collected from species collected from Calliphoridae and Mesembrinellidae in the forest environment (Parque Estadual dos Três Picos—Cachoeiras de Macacu), rural area (UFRRJ—Seropédica), and urban area (UNIRIO, campus Urca—RJ) in the four seasons of the year between 2021 and 2022.

Species	Forest	Urban	Rural	General
Calliphoridae				
*Chrysomya albiceps*	-	Accessory and rare	Accessory and intermediate	Accidental and intermediate
*Chrysomya megacephala*	-	Constant and intermediate	Constant and intermediate	Accessory and intermediate
*Cochliomyia macellaria*	-	-	Constant and intermediate	Accessory and intermediate
*Hemilucilia benoisti*	Constant and intermediate	-	-	Accidental and intermediate
*Hemilucilia segmentaria*	Constant and common	Constant and common	Constant and intermediate	Constant and common
*Hemilucilia semidiaphana*	Constant and common	Constant and intermediate	-	Constant and Common
*Lucilia cuprina*	-	-	Constant and intermediate	Accessory and intermediate
*Lucilia eximia*	Constant and common	Constant and common	Constant and common	Constant and common
*Paralucilia nigrofacialis*	Accessory and intermediate	-	-	Accidental and intermediate
Mesembrinellidae				
*Laneela nigripes*	Constant and common	-	-	Accidental and common
*Mesembrinella bellardiana*	Constant and common	-	-	Accidental and common
*Mesembrinella peregrina*	Accessory and intermediate	-	-	Accidental and intermediate
*Mesembrinella semihyalina*	Constant and intermediate	-	-	Accidental and intermediate
*Mesembrinella currani*	Constant and intermediate	-	-	Accidental and intermediate
*Eumesembrinella quadrilineata*	Constant and intermediate	-	-	Accessory and intermediate
*Eumesembrinella cyaneicyncta*	Accessory and rare	-	-	Accidental and rare
*Eumesembrinella randa*	Constant and rare	-	-	Accessory and intermediate
*Eumesembrinella benoisti*	Accessory and rare	-	-	Accidental and rare
*Huascaromusca aeneiventris*	Accessory and intermediate	-	-	Accidental and intermediate

Constancy index: constant (>50%); accessory (25% > 50%); accidental (<25%). Frequency: rare (1 > 2 individuals); intermediate (3 > 51 individuals); common (>52 individuals).

**Table 4 life-13-01914-t004:** Constancy and absolute and relative frequency of Calliphoridae and Mesembrinellidae species collected from species collected from Calliphoridae and Mesembrinellidae in the forest environment (Parque Estadual dos Três Picos—Cachoeiras de Macacu), rural area (UFRRJ—Seropédica), and urban area (UNIRIO, campus Urca—RJ) in the four seasons of the year between 2021 and 2022.

Constancy of Species	Forest		Rural		Urban	
	N	%	n	%	n	%
Accidental	0	0	0	0	0	0
Accessory	5	33.33	1	20	1	16.67
Constant	10	66.67	4	80	5	83.33
Frequency of species	Forest		Rural		Urban	
	N	%	n	%	n	%
Rare	3	20	1	20	0	0
Intermediate	7	46.67	2	40	5	83.33
Common	5	33.33	2	40	1	16.67

**Table 5 life-13-01914-t005:** Richness, Shannon–Wiener diversity, Simpson dominance, and Pielou evenness of Calliphoridae and Mesembrinellidae species collected in the forest environment (Parque Estadual dos Três Picos—Cachoeiras de Macacu), rural area (UFRRJ—Seropédica), and urban area (UNIRIO, campus Urca—RJ) in the four seasons of the year between 2021 and 2022.

Species	Forest	Rural	Urban
Richness	9	3.50	3.75
Shannon–Wiener diversity	1.77	1.08	1.36
Simpson dominance	0.40	0.56	0.45
Pielou evenness	0.55	0.65	0.71

## Data Availability

The deposited data were made publicly available and can be accessed through: https://doi.org/10.17605/OSF.IO/86NF9.

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
