# Peer review of "Faunistic Analysis of Calliphoridae and Mesembrinellidae (Diptera: Oestroidea) at Different Stages of Bovine Liver Decomposition in the State of Rio de Janeiro"

_life, 2023, doi:10.3390/life13091914_

Round 1

Reviewer 1 Report

I expect some recent reference to family status of Calliphoridae s. lat., with position of modern „families“.

Line 50: A peculiarity of this family among Oestroidea is the larval development that occurs inside the maternal oviduct. ?? (what about Glossinidae?)

l. 116: Simpson Dominance measured the probability that two individuals, randomly .selected from the sample, belong to the same species (nearly all diversity indices are based on the same preposition).

l. 55: Dipterans, as well as other groups of arthropods, characterize… (I do not understand meaning of this sentence).

Part of the text is in another language, e.g.co-ordinates and headline of tab. 1 or: for 48 horas. Line 105.

The collected dipterans were sacrificed in a solution (preserved?)

l. 122: the formula for constancy of occurrence (C) C= n x 100/N is given. Why this formula? It is only percentage of positive samples? (rather primitive for scientific paper). On the other hand, other indices (Shanon, Pielou, Simpson… are not specified and also without any reference.

following taxonomic keys [23, 24]. But these keys are only selection of species, e.g. 24 contains less than ¼ of South American species. In the first paper  (23) names are sometimes wrongly spelled.

Tab3: what means „geral“? Moreover, this column does not alays corresponds with previous.

Shannon-Wiener l. 115, other places in text only Shannon is used (Shannon is correct).

La. nigripes and M. bellardiana. 277 (italics).

l. 289: asynanthropes (not common term, better exoanthropic).

Reference about occurrence of Mesembrinellidae only in exoanthropic environments (humid primary forests) should be added (Guimaraes, 1977 – is already in reference list but not in this sense).

Must be checked, sometimes different lanaguage is used.

Author Response

I expect some recent reference to family status of Calliphoridae s. lat., with position of modern „families“.

Line 50: A peculiarity of this family among Oestroidea is the larval development that occurs inside the maternal oviduct. ?? (what about Glossinidae?)

Reply: The larval development of Mesembrinellidae is different from Glossinidae. In this first family, each larval stage is not well known.

  1. 116: Simpson Dominance measured the probability that two individuals, randomly .selected from the sample, belong to the same species (nearly all diversity indices are based on the same preposition).

Reply: The intention of this sentence is just to explain how Simpson Dominance measured works.

  1. 55: Dipterans, as well as other groups of arthropods, characterize… (I do not understand meaning of this sentence).

Reply: Thanks for that suggestion. I made a small cut in the text in order to try to better elucidate what is said.

Part of the text is in another language, e.g.co-ordinates and headline of tab. 1 or: for 48 horas. Line 105.

Reply: Thanks. It was translated correctly in the text.

The collected dipterans were sacrificed in a solution (preserved?)

Reply: Diptera were sacrificed/killed by the solution. The use of the word "sacrificed" serves to not appear aggressive or offensive to anyone reading it.

  1. 122: the formula for constancy of occurrence (C) C= n x 100/N is given. Why this formula? It is only percentage of positive samples? (rather primitive for scientific paper). On the other hand, other indices (Shanon, Pielou, Simpson… are not specified and also without any reference.

Reply: Apparently this specific formula is less known than the other indices of faunal diversity, such as Shanon, Pielou or Simpson. For this reason, its formula has been described.

following taxonomic keys [23, 24]. But these keys are only selection of species, e.g. 24 contains less than ¼ of South American species. In the first paper  (23) names are sometimes wrongly spelled.

Reply: The aforementioned taxonomic keys were sufficient to be used to identify the species that occurred in this study.

Tab3: what means „geral“? Moreover, this column does not alays corresponds with previous.

Reply: It was a translation error. It's meant general.

Shannon-Wiener l. 115, other places in text only Shannon is used (Shannon is correct).

Reply: This has been corrected in the text.

La. nigripes and M. bellardiana. 277 (italics).

Reply: This has been corrected in the text.

  1. 289: asynanthropes (not common term, better exoanthropic).

Reply: We prefer to use asynanthropes because it presents itself as the exact opposite of synanthropic.

Reference about occurrence of Mesembrinellidae only in exoanthropic environments (humid primary forests) should be added (Guimaraes, 1977 – is already in reference list but not in this sense).

Reply: Reference 29 addresses precisely this.

Reviewer 2 Report

This paper samples four locations in three ecosystems for two families of Diptera using two bait times across several seasons. The data shows what flies come to different baits, and which are common and which are incidental.

I wonder whether this paper would benefit from being rewritten with a focus on forensic entomology, as the ecological angle seems forced. Knowing what insects come to corpses is important for practical forensic entomology: the stuff about "population dynamics, diversity, distribution, dispersal areas and seasonality" does not seem as important, at least to me, nor does your research really address that. The diversity index information also seems relatively useless. We know forests are more diverse than cities.

Mesembrinellidae were found exclusively in the forest. That seems like a significant finding worthy of being stated more clearly in the abstract. The bioindicator angle is good.

Overall the paper is too wordy and would benefit from being written in a less ornate style. Simple sentences expressing clear ideas, with as few words and commas as possible (especially as many grammar errors involving commas were found).

24-27 This line seems to contradict itself. Was there a difference or wasn't there? I recommend rewriting this as two separate sentences that do not use commas: "between the attractiveness in the environments" is not clear.
52 "pupate"
59 "though they are"
102 Are these pitfall traps? If so, state so, else include a diagram or other description of the trap.
105 "hours"
114 delete "which"
138 There's something strange here. A hyperlink? Check the formatting.
140 "adding"
148-149 Same problem as the same line in the abstract. This sentence seems self-contradictory.
151 Why is there a large space here?
152-154 I do not understand this sentence. Can you rewrite it in a way that does not use commas? For example: "Species that occurred only in urban or rural areas were mostly collected in fresh liver."
table 2 Line Hemilucilia segmentaria: There's something off about the formatting here too. A weird, clickable section. Check the pdf.
164 delete "in general"
165 replace "in general" with "in all three locations"
195 It is false that "these organisms are found in both natural and urban environments," as Mesembrinellidae were found exclusively in the forest.
196 Delete "in"
197 What species? if that paper is not about diptera, then don't bother citing it: everybody on earth knows that biological diversity for almost all forms of life is higher in less anthropized areas. It's common sense.
198-207 This reads like a results section. For the discussion, do not use numbers and instead explain what your results mean. What do the ranges of Shannon-Weiner diversity in your study say about the ecosystems? Why should we care? What is meaningful about the diversity being lower in the total area than the urban area (remember that it is not interesting that diversity is lower in urban areas since everyone already expects that… find something interesting to say).
208-218 This would be a good introduction paragraph if you rewrite the paper with a purely forensic direction, to get data on what insects come to carrion in Rio de Janiero for use in future forensic investigations.
209 "subsidies" is the wrong word, but I'm not sure what word you wanted
214 The genera should be italicized
215 "field"
216 Do not use strong language like "fundamental importance"
229-232 italicize the genera
233 "low temporary affluence" seems wrong. Did you mean low temporal abundance?
235 replace "value" with "number"
238-240 It is not necessarily "interesting" that you collected more Calliphorids in fresh bait than putrefied bait, since Calliphoridae are known to be the first colonizers of a corpse, coming to fresh cadavers, not putrefied. That putrefied bait is most commonly used in Diptera studies also does not make your results more interesting: it just suggests the two papers you cited chose a poor bait for Calliphoridae, even if it could have been suitable for other Diptera. You should explain how your results are different from the results of other papers, not from the methods of other papers.
241-243 Now this is interesting! Why did this happen? Was it due to certain species predominating in certain environments, or did the same species present in all 3 environments show this different preference? This needs some explanation.
257 delete "it"
260 italicize the genus
279 "major"
284 delete "for" and italicize the scientific name

Some careless errors (especially regarding italicizing genus names) as well as some translation errors.

Author Response

This paper samples four locations in three ecosystems for two families of Diptera using two bait times across several seasons. The data shows what flies come to different baits, and which are common and which are incidental.

I wonder whether this paper would benefit from being rewritten with a focus on forensic entomology, as the ecological angle seems forced. Knowing what insects come to corpses is important for practical forensic entomology: the stuff about "population dynamics, diversity, distribution, dispersal areas and seasonality" does not seem as important, at least to me, nor does your research really address that. The diversity index information also seems relatively useless. We know forests are more diverse than cities.

Reply: Thank you for the words. We intend to do more articles like this one where we will better address the perspective of forensic entomology by using more types of bait and more stages of decomposition.

Mesembrinellidae were found exclusively in the forest. That seems like a significant finding worthy of being stated more clearly in the abstract. The bioindicator angle is good.

Overall the paper is too wordy and would benefit from being written in a less ornate style. Simple sentences expressing clear ideas, with as few words and commas as possible (especially as many grammar errors involving commas were found).

Reply: Thanks for the suggestions and constructive criticism. It is to be expected that Mesembrinellidae is found only in forested areas. See this article: https://doi.org/10.1590/1519-6984.05614

24-27 This line seems to contradict itself. Was there a difference or wasn't there? I recommend rewriting this as two separate sentences that do not use commas: "between the attractiveness in the environments" is not clear.

Reply: The sentence has been restructured in the text.

52 "pupate"

Reply: Thanks. This has been corrected in the text.

59 "though they are"

Reply: This has been corrected in the text.

102 Are these pitfall traps? If so, state so, else include a diagram or other description of the trap.

Reply: This has been corrected in the text.

105 "hours"

Reply: This has been corrected in the text.

114 delete "which"

Reply: This has been corrected in the text.

138 There's something strange here. A hyperlink? Check the formatting.

I could not see the hyperlink.

140 "adding"

Reply: It didn't seem like the right word to use.

148-149 Same problem as the same line in the abstract. This sentence seems self-contradictory.

151 Why is there a large space here?

Reply: This has been corrected in the text.

152-154 I do not understand this sentence. Can you rewrite it in a way that does not use commas? For example: "Species that occurred only in urban or rural areas were mostly collected in fresh liver."

table 2 Line Hemilucilia segmentaria: There's something off about the formatting here too. A weird, clickable section. Check the pdf.

Reply: I could not see the hyperlink.

164 delete "in general"

Reply: This has been corrected in the text.

165 replace "in general" with "in all three locations"

Reply: This has been corrected in the text.

195 It is false that "these organisms are found in both natural and urban environments," as Mesembrinellidae were found exclusively in the forest.

Reply: This has been corrected in the text.

196 Delete "in"

Reply: This has been corrected in the text.

197 What species? if that paper is not about diptera, then don't bother citing it: everybody on earth knows that biological diversity for almost all forms of life is higher in less anthropized areas. It's common sense.

Reply: This has been corrected in the text.

198-207 This reads like a results section. For the discussion, do not use numbers and instead explain what your results mean. What do the ranges of Shannon-Weiner diversity in your study say about the ecosystems? Why should we care? What is meaningful about the diversity being lower in the total area than the urban area (remember that it is not interesting that diversity is lower in urban areas since everyone already expects that… find something interesting to say).

208-218 This would be a good introduction paragraph if you rewrite the paper with a purely forensic direction, to get data on what insects come to carrion in Rio de Janiero for use in future forensic investigations.

209 "subsidies" is the wrong word, but I'm not sure what word you wanted

Reply: This has been corrected in the text. The word in question was removed but the meaning of the sentence remained.

214 The genera should be italicized

Reply: This has been corrected in the text.

215 "field"

Reply: This has been corrected in the text.

216 Do not use strong language like "fundamental importance"

This has been corrected in the text.

229-232 italicize the genera

Reply: This has been corrected in the text.

233 "low temporary affluence" seems wrong. Did you mean low temporal abundance?

Reply: Precisely that. This has been corrected in the text.

235 replace "value" with "number"

This has been corrected in the text.

238-240 It is not necessarily "interesting" that you collected more Calliphorids in fresh bait than putrefied bait, since Calliphoridae are known to be the first colonizers of a corpse, coming to fresh cadavers, not putrefied. That putrefied bait is most commonly used in Diptera studies also does not make your results more interesting: it just suggests the two papers you cited chose a poor bait for Calliphoridae, even if it could have been suitable for other Diptera. You should explain how your results are different from the results of other papers, not from the methods of other papers.

241-243 Now this is interesting! Why did this happen? Was it due to certain species predominating in certain environments, or did the same species present in all 3 environments show this different preference? This needs some explanation.

Reply: This was a fact that we observed. We cannot deduce much beyond this since more studies with the same focus are needed.

257 delete "it"

Reply: This has been corrected in the text.

260 italicize the genus

Reply: This has been corrected in the text.

279 "major"

Reply: This has been corrected in the text.

284 delete "for" and italicize the scientific name

Reply: This has been corrected in the text.
